# Twelve-year clinical trajectories of multimorbidity in a population of older adults

Davide L. Vetrano [1,2,8 ✉], Albert Roso-Llorach [3,4,8], Sergio Fernández[3,4], Marina Guisado-Clavero[3,4], Concepción Violán[3,4], Graziano Onder[5], Laura Fratiglioni[1,6], Amaia Calderón-Larrañaga[1,9] & Alessandra Marengoni[1,7,9]

Multimorbidity—the co-occurrence of multiple diseases—is associated to poor prognosis, but the scarce knowledge of its development over time hampers the effectiveness of clinical interventions. Here we identify multimorbidity clusters, trace their evolution in older adults, and detect the clinical trajectories and mortality of single individuals as they move among clusters over 12 years. By means of a fuzzy c-means cluster algorithm, we group 2931 people ≥60 years in five clinically meaningful multimorbidity clusters (52%). The remaining 48% are part of an unspecific cluster (i.e. none of the diseases are overrepresented), which greatly fuels other clusters at follow-ups. Clusters contribute differentially to the longitudinal development of other clusters and to mortality. We report that multimorbidity clusters and their trajectories may help identifying homogeneous groups of people with similar needs and prognosis, and assisting clinicians and health care systems in the personalization of clinical interventions and preventive strategies.

[1] Aging Research Center, Department of Neurobiology, Care Sciences and Society, Karolinska Institutet and Stockholm University, Tomtebodavägen 18A, SE-171 65 Solna, Stockholm, Sweden. [2] Centro Medicina dell'Invecchiamento, Fondazione Policlinico Universitario "A. Gemelli" IRCCS, and Università Cattolica del Sacro Cuore, L.go Francesco Vito 1, 00100 Rome, Italy. [3] Fundació Institut Universitari per a la recerca a l'Atenció Primària de Salut Jordi Gol i Gurina (IDIAPJGol), Gran Via 587 àtic, Barcelona, Spain. [4] Universitat Autònoma de Barcelona, Campus de la UAB, Plaça Cívica, 08193 Bellaterra (Cerdanyola del Vallès), Spain. [5] Department of Cardiovascular, Endocrine-Metabolic Diseases and Aging, Istituto Superiore di Sanità, Via Giano della Bella 34, 00161 Rome, Italy. [6] Stockholm Gerontology Research Center, Sveavägen 155, 11346 Stockholm, Sweden. [7] Department of Clinical and Experimental Sciences, University of Brescia, Piazza Mercato 15, 25121 Brescia, Italy. [8] These authors contributed equally: Davide L Vetrano, Albert Roso-Llorach. [9] These authors jointly supervised this work: Amaia Calderón-Larrañaga, Alessandra Marengoni. ✉email: davide.vetrano@ki.se

As people age they tend to develop multiple chronic diseases; the term multimorbidity identifies this condition[1]. After 60 years of age, 55–98% of people are affected by two or more chronic diseases, and patients with multimorbidity account for up to 80% of consultations with general practitioners and virtually all consultations with geriatricians[2,3]. Co-occurring diseases interact with each other, increasing the risk of negative events beyond the sum of the risk of each disease[4]. Multimorbidity triggers complex pharmacological regimes, increases the use of health care resources, and reduces the quality and length of life[1,4–6]. It challenges physicians, who are usually trained to consider only a limited number of interactions between diseases and between diseases and drugs, and it puts pressure on health care systems, which struggle to offer older adults with multimorbidity comprehensive assessment, effective treatments, and integrated care paths[6–10]. Moreover, because older people with multimorbidity are usually excluded from randomized clinical trials, there are few clear recommendations about how to provide health care for older adults with multimorbidity. Complexity is thus translated into frustrating uncertainty and powerlessness and affects the quality of care at every level of the health care process[9].

Both clinical experience and epidemiological studies suggest that diseases cluster in the same person according to specific patterns[5,11]. Several clusters of diseases have been identified with some consistency across studies; however, there are a number of discrepancies in study findings[12]. A systematic review by Prados-Torres et al. identified 97 clusters of multimorbidity, and the findings of most of the reviewed studies suggested three clusters of multimorbidity: cardiometabolic, mental health, and musculoskeletal. At the same time, the studies in the review identified many unexplained heterogeneous clusters[12]. In addition to between-study methodological differences, one of the explanations for this finding may lie in the dynamic nature of disease clusters, which is not accounted for in cross-sectional studies. These clusters evolve overtime, and mortality selection plays an important role in shaping the observed population[13]. Capturing such dynamism is the only way to better understand the natural history of multimorbidity and to shed light on previously unexplained findings.

Most previous studies in this field have focused on clusters from the viewpoint of disease analyses rather than the analysis of groups of individuals[12,14]. Focusing on people is in keeping with the principle of patient-centered care and can provide information that facilitates the move toward personalized medicine[15]. A better understanding of older adults' transitions among multimorbidity clusters overtime may help detect homogeneous groups of individuals who may benefit from similar preventive (secondary and tertiary) interventions, treatment, and care. We therefore aimed to identify multimorbidity clusters in a population-based cohort of older adults, trace the evolution of the clusters over 12 years, and follow the clinical trajectories of the individuals as they moved between clusters or to death over time.

We found that multimorbidity clusters change dynamically overtime in older adults, following different clinical trajectories. Different clusters are also associated with different prognosis. Multimorbidity trajectories may help identifying homogeneous groups of people with similar needs and prognosis, and assisting clinicians and health care systems in the personalization of clinical interventions and preventive strategies.

## Results

### Six clusters of individuals with multimorbidity were identified.

The present study is based on data from the Swedish National Study on Aging and Care in Kungsholmen (SNAC-K), an ongoing population-based study started in 2001 and involving 3363 individuals aged ≥60 years from a central area in Stockholm, Sweden. From the whole sample, 432 participants with <2 chronic disease have been excluded (i.e., those without multimorbidity). Those excluded were younger, reported a higher level of education, and were more often male than those included in the study (p for t test < 0.001). At baseline, study participants' mean age was 76.1 ± 11.0 [standard deviation] and 1951 (66.6%) were female. Over the 12 years, 1290 (44%) deaths occurred (28% within the first 6 years and 16% between 6 and 12 years). Moreover, 625 (22%) individuals dropped out (14% within the first 6 years and 8% between 6 and 12 years). At each follow-up, we performed a dimensionality reduction (i.e., multiple correspondence analysis) to obtain the input data for participants' clustering. A fuzzy c-means cluster analysis with optimal a fuzziness parameter at $m = 1.1$ (which outperformed other $m$ values; see "Methods") was employed to identify clusters of individuals based on their underlying patterns of multimorbidity. Using an observed/expected ratio ≥2 (O/E ratio; i.e., the ratio between the prevalence of a given condition in a cluster and its prevalence in the whole sample) and an exclusivity ≥25% (i.e., the proportion of individuals with a given condition in the whole sample that belong to a cluster) for each disease, five clusters of people were identified at baseline: those with *psychiatric and respiratory diseases* (5.4%), *heart diseases* (9.3%), *respiratory and musculoskeletal diseases* (15.7%), *cognitive and sensory impairment* (10.6%), and *eye diseases and cancer* (10.7%). Solutions were evaluated based on their clinical consistency and significance criteria (Supplementary Figs. 1–15). Half of the people (48.7%) were grouped in an additional *unspecific* cluster, as they were affected by prevalent diseases but whose occurrence did not exceed the expected. Similarly, five clusters (plus the unspecific one) were identified at 6 and 12 years. At follow-ups, those diseases characterizing the baseline clusters were regrouped into different multimorbidity clusters. The clinical characteristics of the clusters are reported in Supplementary Table 1.

**Individuals had different demographic, clinical and functional profiles across the clusters.** Descriptive analyses were carried out to characterize the six clusters of individuals with multimorbidity. At baseline, participants in the *cognitive and sensory diseases*, the *eye diseases and cancer*, and the *heart diseases* clusters were the oldest. Participants in the *heart diseases*, the *eye diseases and cancer*, and the *psychiatric and respiratory diseases* clusters presented the greatest number of chronic diseases (mean number: 7.7 ± 2.4 [standard deviation], 6.0 ± 2.0, and 5.7 ± 2.2, respectively). Participants in the *heart diseases* and *psychiatric and respiratory diseases* clusters and those in the *cognitive and sensory impairment* cluster used the highest number of drugs (mean number: 7.7 ± 3.5, 6.2 ± 3.7, and 6.1 ± 3.4, respectively). Moreover, individuals included in the *heart diseases*, the *eye diseases and cancer*, and the *cognitive and sensory impairment* clusters presented the highest prevalence of disability and slow walking speed. The *cognitive and sensory impairment* and the *psychiatric and respiratory diseases* cluster showed the lowest Mini-Mental State Examination (MMSE) scores. The *unspecific* cluster was characterized by the lowest mean age and the lowest number of chronic diseases and drugs. This group had the lowest prevalence of disability and the highest walking speed, yet it had a high prevalence of hypertension, diabetes, dyslipidemia, and obesity. Such conditions were frequent also among participants in the *heart diseases* and the *eye diseases and cancer* clusters.

At follow-ups, in spite of varied clustering, a similar clinical distribution was observed for the different types of disorders. That is, people in clusters characterized by cardiovascular,

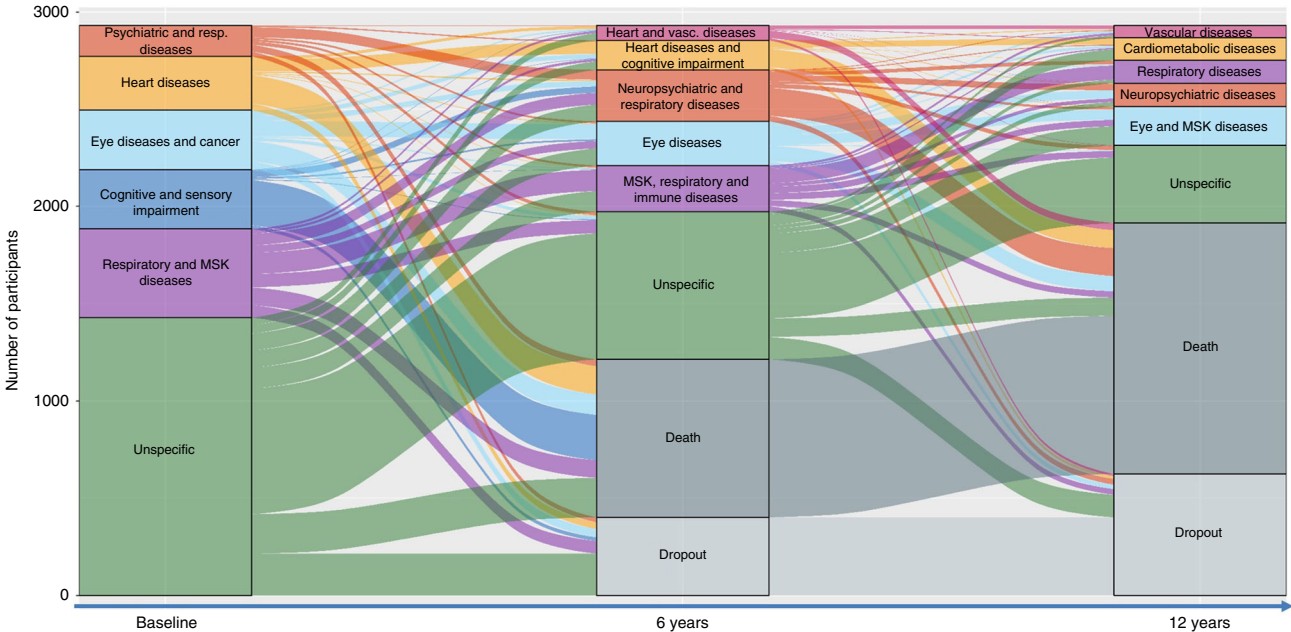

**Fig. 1 Evolution of multimorbidity clusters and clinical trajectories of older adults with multimorbidity over 12 years.** The height of the boxes and the thickness of the stripes are proportional to the amount of people belonging to the cluster and moving from the cluster, respectively. MSK musculoskeletal. To note, for this analysis participants were assigned to the cluster they were more likely to belong in order to investigate the most likely trajectories.

neuropsychiatric, and respiratory diseases showed the highest number of diseases and drugs and the highest levels of functional impairment.

**Patterns of transitions between clusters can be identified over time.** Upon assigning the individuals into the cluster they were more likely to belong to, we described their trajectories as they moved between clusters or to death over time. Figure 1 depicts the longitudinal evolution of multimorbidity clusters over 12 years and includes both the change overtime of disease patterns (the diseases that characterize a specific cluster of individuals) and the migration of participants from one cluster to another. The height of the boxes and the thickness of the stripes in the figure are proportional to the amounts of people in the cluster and moving out from the cluster, respectively.

In order to better characterize such transitions, we report in Figs. 2 and 3 the proportion of participants that were part of the 6-year and 12-year follow-ups clusters and that moved from multimorbidity clusters detected at an earlier wave. The percentages of participants moving from baseline and 6-year clusters, to 6-year and 12-year clusters, respectively, are reported in Supplementary Tables 4–7. During both first and second follow-up periods, the main shifts among clusters involved participants in the *unspecific* cluster, who moved primarily to clusters characterized by cardiovascular, eye, respiratory, and musculoskeletal diseases. For example, persons in the *unspecific* group at baseline moved and represented 48.7%, 45.0%, and 38.8% of the 6-year follow-up *heart and vascular diseases*, *musculoskeletal, respiratory and immune diseases*, and *eye diseases* clusters, respectively. Similarly, persons belonging to the *unspecific* group at the 6-year follow-up moved and represented 49.5%, 49.1%, and 20.6% of the 12-year follow up *cardiometabolic diseases*, *eye and musculoskeletal diseases*, and *vascular diseases* clusters, respectively.

**Different multimorbidity clusters confer different mortality risks.** The association between the multimorbidity clusters and mortality was tested in logistic regression models adjusted by age,

sex, and education, taking the *unspecific* cluster as the reference group. As shown in Table 1, at baseline the *heart diseases* (OR 3.07; 95% CI 2.26–4.19), the *cognitive and sensory impairment* (OR 6.00; 95% CI 4.21–8.54), and the *psychiatric and respiratory diseases* (OR 1.60; 95% CI 1.02–2.51) clusters were significantly associated with a higher six-year mortality, compared with the people in the *unspecific* cluster. These clusters accounted for 51% of deaths. At first follow-up, the *heart and vascular diseases* (OR 3.78; 95% CI 2.13–6.70), the *heart diseases and cognitive impairment* (OR 3.73; 95% CI 2.41–5.79), and *neuropsychiatric and respiratory diseases* (OR 4.30; 95% CI 2.95–6.27) clusters had the highest OR for 6-year mortality, compared with the group of people in the *unspecific* cluster. These clusters accounted for 57% of deaths in the following 6 years.

## Discussion

Tracing the evolution of multimorbidity clusters and the clinical trajectories of older adults with multimorbidity overtime led to two major findings. The first was a high heterogeneity in the multimorbidity clustering at baseline. Only half of the participants could be grouped into a well-characterized cluster: *psychiatric and respiratory diseases, heart diseases, respiratory and musculoskeletal diseases, cognitive and sensory impairment*, and *eye diseases and cancer*. The other half of the participants were sorted into an unspecific cluster and were characterized by having a younger age, lower numbers of co-occurring diseases and drugs, good functional and cognitive abilities, and a high percentage of cardiovascular risk factors. The second major finding was a highly dynamic evolution of multimorbidity clusters at both 6 and 12 years. Over 12 years, changes in cluster composition, participants' transitions from one cluster to another, and participant mortality generated a dynamic but well-defined clinical picture. The first remarkable trajectory involved the group of people part of the unspecific cluster at baseline. The number of participants grouped in this cluster halved at the 6- and 12-year follow-ups as the majority transitioned toward the specific multimorbidity clusters identified at follow-ups. Given the young age and less complex clinical picture of these individuals, they may be considered an

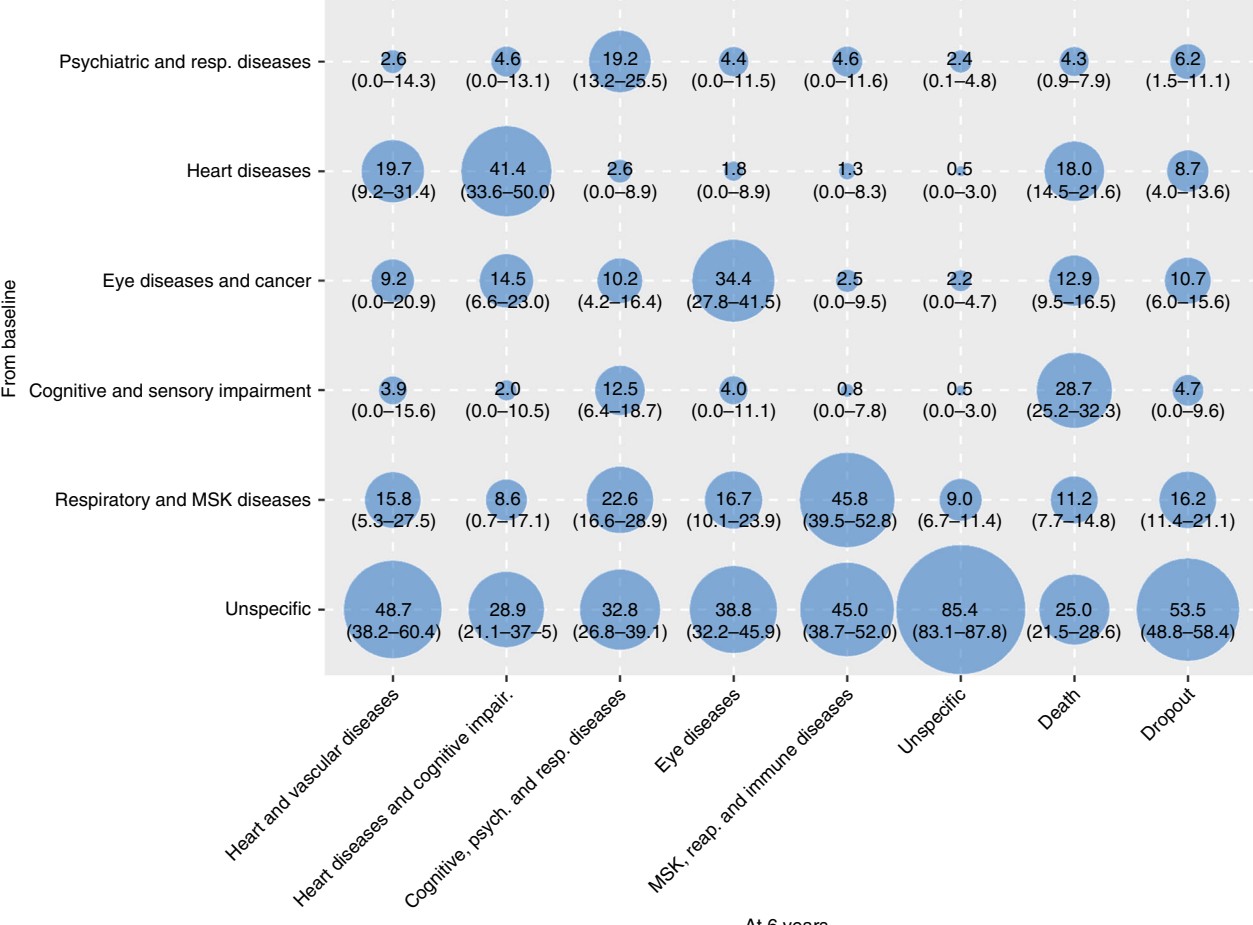

**Fig. 2 Contribution of the baseline multimorbidity clusters to the 6-year follow-up clusters.** Numbers indicate the percentage (%) of people belonging to the 6-year follow-up clusters that moved from the baseline clusters. To note, for this analysis participants were assigned to the cluster they were more likely to belong.

at-risk population for developing more complex multimorbidity and as such potentially susceptible to preventive intervention. The second relevant trajectory was the high mortality of individuals in clusters characterized by cardiovascular and neuropsychiatric diseases, which, despite representing only 25%, 28%, and 29% of the participants at baseline, 6 years, and 12 years, respectively, accounted for 51% and 57% of deaths during the first and second follow-up periods, respectively.

Increasingly, studies are analyzing clusters of multimorbidity across different populations, settings, and countries, but most studies have had a cross-sectional design or focused on the progression of co-morbidities of index diseases[12,16,17]. There is scanty evidence of how clusters of multimorbidity change over-time. The comparison is also limited by the fact that previous studies have used primary care, hospital-based registries or self-reported diagnoses, included only middle-aged people, or examined both acute and chronic conditions. A study from Spain that used a similar analytical strategy on large data from electronic primary health care records identified six clusters of multi-morbidity: musculoskeletal, endocrine-metabolic, digestive/respiratory, neuropsychiatric, cardiovascular, and an unspecific group. These clusters exhibited less variation during the 6 years of follow-up than the patterns identified in our study, which could be explained by our longer follow-up period[18]. The use of electronic health records may also have led to an under detection of less severe diseases and multimorbidity[19]. A study from the Netherlands focused on six cardiovascular conditions. Clinical

data from a large sample of general practice showed that the more diseases present at baseline, the higher the cumulative incidence rates of one or more new diseases (up to 47% at the 3-year follow-up and 76% at the 5-year follow-up)[20]. Another study of a population-wide registry of more than six million patients in Denmark showed more than a thousand significant longitudinal disease trajectories and some major multimorbidity clusters characterized by diseases of the prostate, chronic obstructive pulmonary disease, cerebrovascular disease, cardiovascular disease, and diabetes mellitus. The study had the limitation of data drawn retrospectively from a hospital registry of primary and secondary diagnostic codes. Further, both chronic and acute diseases were included[21], making the findings difficult to compare with ours. Finally, in an Australian study more than 13,000 middle-aged women with no history of diabetes, heart disease, or stroke at baseline were followed for 20 years in order to evaluate the longitudinal progression of the three conditions. Over 20 years, 18% of the women progressed to at least one condition, and 16.8% had two or three of these conditions; moreover, the onset of stroke was more strongly associated with an increased risk of progressing to the other two diseases. This is in contrast with what we observed in our study, which showed an opposite transition, from cardiovascular risk factors (e.g., diabetes) to overt cardiovascular and neuropsychiatric diseases. In the same Australian study, social inequality, obesity, hypertension, physical inactivity, smoking, and other chronic conditions were significantly associated with the three diseases independently but

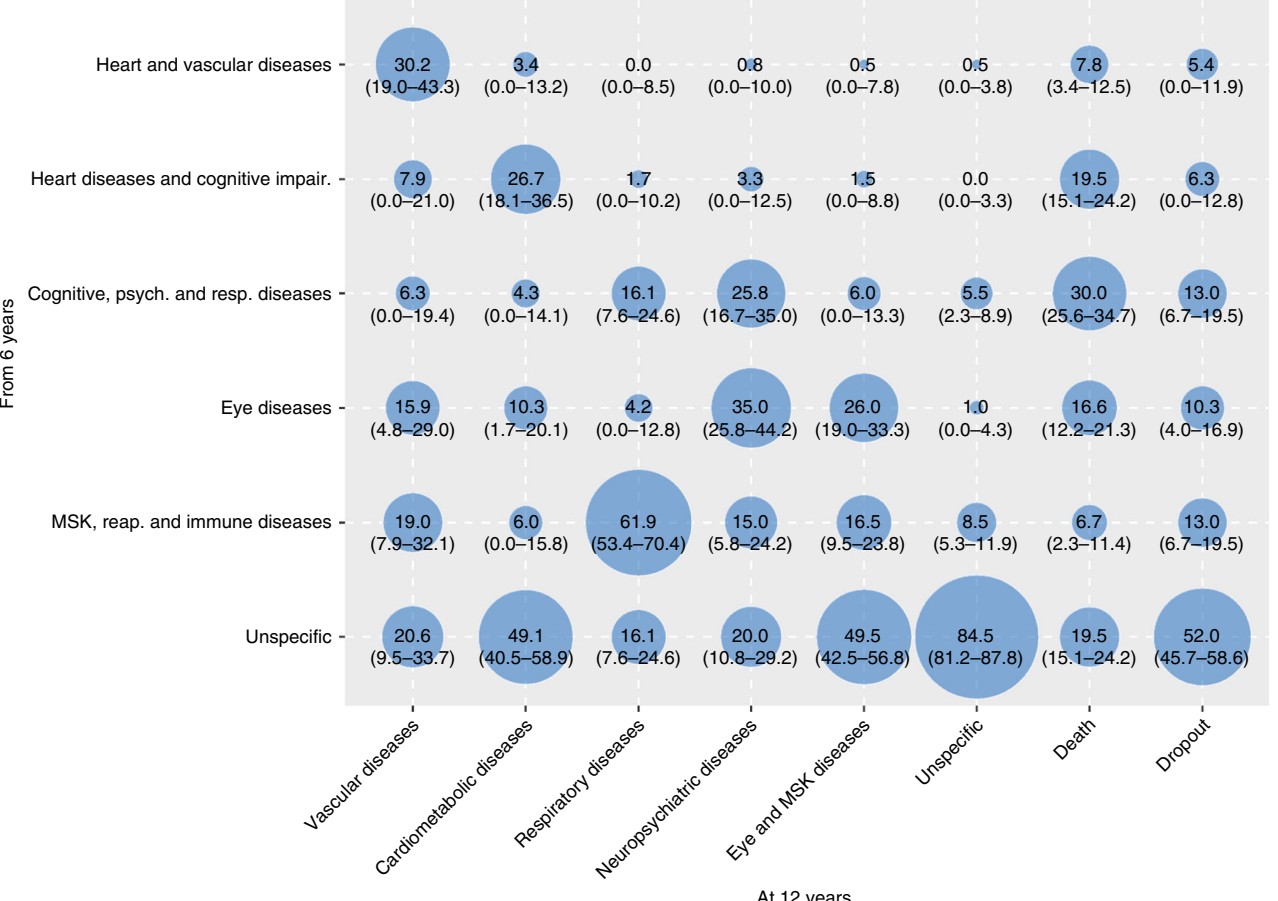

**Fig. 3 Contribution of the 6-year follow-up multimorbidity clusters to the 12-year follow-up clusters.** Numbers indicate the percentage (%) of people belonging to the 12-year follow-up clusters that moved from the 6-year follow-up clusters. To note, for this analysis participants were assigned to the cluster they were more likely to belong.

also with their co-occurrence. The study used self-reported diagnoses[22].

Some diseases may not be as independent of each other as we have previously thought. Biological, health-care related (e.g., pharmacological treatment), and psychosocial factors may increase susceptibility to a specific disease or to diseases in general in an individual[1,23]. Such factors can systematically drive diseases clustering beyond chance as well as their evolution to other clusters over time. First, direct consequences may explain why a large number of people in the *heart diseases* cluster at baseline became part of the *heart diseases and cognitive impairment* cluster at 6 years. Extensive scientific evidence supports the association between heart disease and cognitive decline through different mechanisms such as emboli, ischemic events, small vessel disease, cerebral hypoperfusion, and hypoxia. Indeed, mixed dementia, resulting from both cerebrovascular lesions and neurodegeneration, accounts for the majority of dementia cases among very old individuals[24]. Second, treatment consequences are another possible pathway when a disease occurs as the result of the pharmacological or surgical treatment of another condition. For example, part of the *neuropsychiatric and respiratory diseases* cluster, an association that remained over the entire course of our study, may be linked to the steroid treatment of respiratory diseases. Steroid treatment can often cause neurotic disorders and depression[25]. Third, overlapping symptomatology may result in diseases being misdiagnosed in an initial phase. This may have occurred with some baseline psychiatric conditions in the *psychiatric and respiratory diseases* cluster, which by 6 or 12 years

may have evolved into, or been correctly classified as, cognitive impairment and dementia, putting them in the *cognitive impairment, psychiatric and respiratory diseases* cluster.

Finally, the *unspecific* cluster deserves special attention. These participants were characterized by diseases that were not over-represented. However, despite their younger age and better physical and mental fitness, they had a high prevalence of cardiovascular and metabolic risk factors (diabetes, obesity, dyslipidemia, and hypertension). At baseline, almost half of the sample was part of this group. These people contributed to 29–49% of the multimorbidity clusters at the 6-year follow-up and to 16–50% of the multimorbidity clusters at 12 years, especially to those characterized by cardiovascular, eye, respiratory, and musculoskeletal diseases. Despite it is now well established that cardiometabolic conditions such as diabetes, obesity, dyslipidemia, and hypertension are important risk factors for the development of several cardiovascular diseases, less is known about the same risk factors, and the risk of other chronic conditions[26,27]. A few individuals moved from a specific cluster to the unspecific cluster over time. This may be explained by the fact that the progressive accrual of new diseases and the mortality (or dropout) of participants included in any of the specific clusters changed the reciprocal relation among diseases in survivors—in terms of prevalence, O/E ratio and exclusivity—making some of the subjects no longer classifiable into a specific cluster.

At least four out of ten participants died over the course of the study. Both at baseline and at 6-year follow-up, individuals with multimorbidity patterns characterized by cardiovascular and

**Table 1 Association between clusters and mortality during the first (0–6 years) and second (6–12 years) follow-up.**

| Multimorbidity clusters at baseline | Events/at risk (%) | OR (95% CI)* 0–6 years mortality | Multimorbidity clusters at 6 years | Events/at risk | OR (95% CI)* 6–12 years mortality |
|---|---|---|---|---|---|
| Psychiatric and respiratory diseases | 35/159 (22) | 1.60 (1.02–2.51) | Heart and vascular diseases | 37/76 (49) | 3.78 (2.13–6.70) |
| Heart diseases | 146/277 (53) | 3.07 (2.26–4.19) | Heart diseases and cognitive impairment | 93/152 (61) | 3.73 (2.41–5.79) |
| Eye diseases and cancer | 105/305 (34) | 1.23 (0.90–1.68) | Neuropsychiatric and respiratory dis. | 143/265 (54) | 4.30 (2.95–6.27) |
| Cognitive and sensory impair. | 233/306 (76) | 6.00 (4.21–8.54) | Eye diseases | 79/227 (35) | 1.33 (0.89–2.00) |
| Respiratory and MSK diseases | 91/456 (20) | 1.29 (0.96–1.74) | MSK, respiratory, and immune diseases | 32/238 (13) | 1.06 (0.67–1.70) |
| Unspecific group | 203/1428 (14) | Ref. | Unspecific group | 93/758 (12) | Ref. |

To note, for this analysis participants were assigned to the cluster they were more likely to belong.
Asterisk adjusted for age, sex, and education.
OR odds ratio; CI confidence interval; MSK musculoskeletal.

neuropsychiatric diseases had the highest mortality; with adjusted odds ratios ranging between 1.60 and 6.00 (taking people in the *unspecified* cluster as the reference). Those clusters accounted for 51% of deaths during the first follow-up and for 57% of deaths during the second follow-up. Notably, at 6 years there were two clusters characterized by cardiovascular diseases. Cardiovascular and neuropsychiatric diseases—the former including diseases such as heart failure and coronary diseases and the latter including diseases such as dementia and depression—are frequent and burdensome chronic conditions in older adults and are among the most important determinants of years of life spent with disability[28]. This is in line with a previous study from our group, showing that neuropsychiatric disease clusters, especially when combined with one or multiple cardiovascular diseases, have the highest impact on function decline in older persons[5]. Such findings were confirmed in other studies as well[29–31]. Indeed, the high mortality of people belonging to neuropsychiatric and heart disease clusters was not surprising as those clusters had the highest functional disability and lowest walking speed both at baseline and at first follow-up. Similar findings were reported also in studies from Spain[13] and from the United Kingdom[4]. The authors of the first report found that, compared with those subjects part of the musculoskeletal cluster, women in the cardiovascular clusters had the highest risk of dying. In the latter, co-occurring cardiometabolic disorders, unlike single disorders, decreased survival in a multiplicative way. It can be argued that not all diseases included in the cardiovascular or neuropsychiatric clusters transmit the same mortality risk. In fact, the nature of diseases, their impact at the organism level, and their severity may play major prognostic roles[13]. However, previous studies conducted in the field of associative multimorbidity have shown that the group-specific effect of clusters of diseases remains regardless of the role played by single diseases[5].

The main strength of this study was the thorough clinical evaluation that underlay disease assessment. Each participant in SNAC-K undergoes a 5 h comprehensive assessment that follows a standard protocol and is carried out by a physician, a nurse, and a psychologist. We then categorized diseases using a strict clinically driven method developed and tested by our group[32]. Furthermore, the lack of missing information on disease status increases the internal validity of our study. Another major strength of this study was the statistical method, which allowed us to cluster people by their co-occurring diseases. We took advantage of the method to follow individuals overtime and track their trajectories. The fuzzy c-means cluster algorithm is the choice method for pattern recognition when clusters tend to overlap, which is often the case as older adults present high prevalence of co-occurring conditions. In contrast to previous studies, each participant was assigned a probability of belonging to a cluster without being forced to be part of it exclusively. Other strengths included the long follow-up time, the high number of very old people, and the large age span of the participants (60–104 years). Moreover, including both mental and physical conditions in the analyses gave us the opportunity to investigate the interplay, potentially bidirectional, between mental health problems and chronic physical conditions. Several limitations of the present study should be mentioned. First, diseases were considered regardless their severity. Disease severity may indeed partially explain the clinical trajectories described in the present study. However, the interaction among different comorbidities still seems to play a major role—as it has been shown by us and others in previous studies—even when measures of disease severity are taken into account[4,5,31,33]. Moreover, in our opinion, independently from disease severity, the insights on the natural evolution of multimorbidity provided in this study are highly valuable and cover an important knowledge gap left by previous

cross-sectional studies. Further, there is evidence that the burden of specific conditions changes depending on the overall multimorbidity status of one individual, making it difficult—especially in older individuals—to ascertain the relevance of single disease severity. Second, the dropout rate of participants (14% at 6 years and 8% at 12 years) may have affected cluster definition. However, to the best of our knowledge, this is an exceptionally low figure compared with studies of this type. Third, the discontinuous follow-up carried out in SNAC-K—every 3 or 6 years—may have affected disease detection and consequently the cluster analysis, especially among people who died or dropped out during the observation period. Finally, the average high socio-economic status of participants in SNAC-K may potentially limit the generalizability of the findings.

Over their life course, individuals develop multiple diseases. This challenges the current organization of medical care services and the traditional research approach based on single diseases. Programs that bridge multiple clinical specialties and health care units should be developed to focus on single individuals, their specific clinical profiles, and their specific clinical trajectories[34]. Knowing how diseases cluster together, and importantly, how the clinical status of people with multimorbidity can change over subsequent years helps not only in understanding the complexity and dynamic evolution of multimorbidity clusters but also in supporting clinicians who manage co-occurring chronic diseases and health policy makers who plan care resources use. The findings from our study contribute in many ways. Firstly, they help identify people at high risk of progressing to severe disease clusters with worse prognosis. The people who could not be grouped in any specific cluster are at risk of cumulating further chronic disorders and increasing the severity of their multimorbidity profile. However, 28% of the people in this group remained relatively healthy during follow-ups. They had the lowest numbers of co-occurring chronic diseases and drugs and a better functional status than people in specific multimorbidity clusters, providing a large time window for preventive intervention. Future studies should focus on promotion of healthy aging in this group of individuals. Our findings contribute secondly to the development of personalized medicine in multimorbidity as our analysis is based on individuals and not diseases. There is solid evidence that persons who are affected by multimorbidity, face complex treatments, and require continuous monitoring far better from primary care with a patient-centered approach[35]. The strong transition we found from heart to brain diseases gives impetus to efforts in primary care to treat and monitor patients affected by heart disease. Treatment adherence is very low among older people with multimorbidity and heart diseases in particular[36]. Thirdly, our findings support prognostic counseling for patients and caregivers, given the high mortality of people with co-occurring mental and cardiovascular disorders. Fourthly, our findings encourage the planning of future randomized clinical trials toward the better management of multimorbidity. The 3D approach recently proposed by Salisbury et al. is an example of an intervention that could have focused on those multimorbidity clusters that may most likely lead to negative health outcomes (neuropsychiatric and cardiovascular clusters)[37]. In this pragmatic trial, the target population was selected based exclusively on the number of diseases and did not take into account specific groups of diseases. This may explain why the intervention was not able to improve participants' quality of life[38].

In conclusion, clinical trajectories of older adults with multimorbidity are characterized by great dynamism and complexity but can still be tracked over time. By analyzing data from a large population-based study of people aged 60+ years, we were able to identify multimorbidity clusters, trace their evolution overtime, and follow individuals' trajectories over 12 years. Shared risk factors and

pathophysiology, development of diseases as a consequence of other conditions or treatments, and symptomatic overlap among diseases underlie most of the trajectories identified. Although the ability to discriminate among the potential mechanisms underlying the co-occurrence of multiple chronic diseases needs further improvement, taking into account multimorbidity clusters, and their evolution overtime may enable better decisions for patients with multimorbidity at every health care level and better tailoring of the target population in future interventions.

## Methods

**Study population.** We used longitudinal data from the population-based SNAC-K[39]. The study population consists of adults ≥60 years living in the community or in institutions in the Kungsholmen district of Stockholm, Sweden. A random sample of 11 age cohorts born between 1892 and 1939 (the youngest and oldest age cohorts were oversampled) was invited to participate in the study. People who agreed to participate were evaluated for the first time between 2001 and 2004. Participants who were <78 years of age were then followed up every 6 years and participants ≥78 years every three years. The present study is based on data collected at baseline, 6 years, and 12 years. At baseline, 3363 people were examined (participation rate 73%). Overall, 432 participants were excluded because they did not have multimorbidity (≥2 chronic diseases) at baseline. The study was approved by the Regional Ethics Review Board in Stockholm. Participants in the study provided written informed consent. For participants with prevalent or incident cognitive impairment, written informed consent was obtained from the next of kin. The present study was reported in keeping with the STrengthening the Reporting of OBservational studies in Epidemiology recommendations.

**Chronic diseases.** At each study wave, SNAC-K participants undergo an ~5 h-long comprehensive clinical and functional assessment carried out by trained physicians, nurses, and neuropsychologists. Physicians collect information on diagnoses via physical examination, medical history, examination of medical charts, self-reported information, and/or proxy interviews. Clinical parameters, lab tests, drug information, and inpatient and outpatient care data are also used to identify specific conditions. All diagnoses are coded in accordance with the International Classification of Diseases, 10th revision (ICD-10). In the current study we sorted the ICD-10 codes into 60 chronic disease categories in accordance with a clinically driven methodology (Tables S2 and S3)[32]. To avoid statistical noise and the resulting spurious findings in the models, we excluded diseases with a prevalence of <2%. In SNAC-K at each study wave, drugs are coded in accordance with the Anatomical Therapeutic Chemical classification.

**Vital status and loss to follow-up.** Information about vital status was derived from death certificates provided by Statistics Sweden, the Swedish governmental statistics agency. Survival status was assessed throughout the follow-up period. Participants were considered lost to follow up if they or a proxy declined to participate, could not be contacted, had moved out of the study area, or canceled an assessment.

**Other variables.** Information on demographics (age, sex, and education) was collected during nurse interviews. We divided education into elementary, secondary, university, or higher. Level of disability was measured as the sum of the basic and instrumental activities of daily living (ADL and IADL) a person was unable to perform independently[40]. The six ADLs were bathing, dressing, toileting, continence, transferring, and eating. The eight IADLs were grocery shopping, meal preparation, housekeeping, doing laundry, managing money, using the telephone, taking medications, and using public transportation. Walking speed (m/s) was assessed by asking participants to walk 6 m at their usual speed or 2.44 m if the participant reported walking quite slowly. Speeds of <0.8 m/s were categorized as impaired[41]. Cognitive status was assessed by physicians using the MMSE, with a score range of 30 at best to 0 at worst[42].

**Statistical analysis.** Sample characteristics at baseline, 6-year follow-up, and 12-year follow-up were described for each multimorbidity cluster using weighted means and proportions obtained by the membership matrix (see below). At each study wave, clusters of older adults who shared patterns of multimorbidity were independently identified using the fuzzy c-means cluster analysis algorithm, which belongs to the family of *soft* clustering algorithms. The algorithm estimates $c$ cluster centers (similar to $k$-means) but with fuzziness so that individuals may belong to more than one cluster. The use of a fuzzy cluster analysis over a hard cluster analysis helps to better handle the stochastic nature of some disease association, the potential noise stemming from the measurement (e.g., disease assessment), and the variance due to between-individual differences. Through this technique, we obtained clusters of individuals and a membership matrix that indicated the degree of participation of each subject in each cluster. In a second step, to evaluate the most likely clinical trajectories of the participants as they moved among clusters

over time, each individual was assigned to the cluster with the highest membership score at each time point. We used dimensionality reduction techniques (multiple correspondence analysis) to obtain the input data for clustering the participants. The Karlis–Saporta–Spinaki rule was used to decide how many dimensions to retain[43]. The main parameters used during our cluster analysis were the number of clusters and a fuzziness parameter, denoted as "$m$", which ranges from just above 1 to infinity. High $m$ values produce a fuzzy set of $c$ clusters, so that individuals are equally distributed across clusters, whereas lower ones generate non-overlapped clusters. In our study we checked $m = 1.1, 1.2, 1.4, 1.5, 2, 4$ over 1 to 20 cluster combinations; the value $m = 1.1$ over performed the rest of values. Since clustering algorithms are unsupervised techniques, the model fitting the dataset is traditionally computed through cost functions that depend on both the dataset and the clustering parameters and are denoted as validation indices. We computed different validation indices to obtain the optimal number of clusters $c$ and the optimal value of the fuzziness parameter $m$. Included parameters were: the Fukuyama index (optimal when presenting low values), the Xie–Beni index (optimal when presenting low values), the Partition coefficient index (optimal when presenting high values), the Partition entropy index (optimal when presenting low values), and the Calinski–Harabasz index (optimal when presenting high values; Supplementary Figs. 1–15)[44]. Given the stochastic nature of the clusters, we ran 100 independent clustering repetitions to obtain the average final solution. We based our evaluation of the consistency and significance of the final solution on clinical criteria. To cross-validate the model, we randomly split the individuals into two independent data sets and compared their validation indices. Indices were computed and averaged over 100 repetitions. To characterize the clusters of multimorbidity that corresponded to each cluster of individuals, we calculated the frequency of chronic diseases in each cluster. Observed/expected ratios ($O/E$-ratios) were calculated by dividing the prevalence of a given disease within a cluster by its prevalence in the overall population. The exclusivity of different diseases, defined as the fraction of participants with the disease in the cluster over the total number of participants with the disease, was also calculated. We considered a disease to be associated with a given cluster of individuals when the $O/E$ ratio was ≥2 or the exclusivity was ≥25%[18]. Such criteria were used to name multimorbidity clusters after the diseases that mostly characterized them. To note, due to the dynamism of the phenomenon, the names of the clusters change overtime, reflecting the evolving combinations of diseases that characterize them at each time point. Shifts between clusters were computed by cross-tabulating individuals between each wave (baseline to 6-year follow-up and 6-year to 12-year follow-up) after assigning them individuals to the cluster where they were more likely to belong. In this way, we analyzed the most likely individual trajectories. Frequencies (percentages) of participants who changed from one cluster to another were computed to assess the overlap between waves. Both column percentages and row percentages are provided in Supplementary Tables. Mortality and dropout status were considered as fixed clusters in both 6-year and 12-year follow-ups. Logistic regression models adjusted by age, sex and education were fitted to estimate the association between clusters and mortality, using the *unspecific* cluster as the reference group. Also in this case, participants were assigned to the cluster where they were more likely to belong. Odd ratios (OR) and 95% confidence intervals (CI) were adjusted for age, sex, and education. All comparisons were adjusted for multiplicity. Pairwise comparison of $p$ values, corrected for multiple comparisons, was calculated. Tukey method were used when the explanatory variable was normal-distributed or Benjamini and Hochberg method otherwise[45]. The significance level was set at $p = 0.05$. Although the overall number of significant tests between clusters at each follow-up remained stable at each follow-up, the number of highly significant pairwise statistical test (i.e., $p < 0.001$) decreased from 60.0 to 36.7%. Statistical analyses were performed using R 3.5.1 and Stata 15. Codes are available on demand.

**Reporting summary**. Further information on research design is available in the Nature Research Reporting Summary linked to this article.

## Data availability
The source data underlying all the figures and tables (including supplementary ones) is represented by the SNAC-K project, a population-based study on aging and dementia (http://www.snac-k.se/). Access to these original data is available to the research community upon approval by the SNAC-K data management and maintenance committee. Applications for accessing these data can be submitted to Maria Wahlberg (Maria.Wahlberg@ki.se) at the Aging Research Center, Karolinska Institutet.

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

## Acknowledgements

We thank the SNAC-K participants and the SNAC-K Group for their collaboration in data collection and management, and scientific editors Kimberly Kane and Karen Hagersten for useful comments on the text. This work was supported by the funders of the Swedish National study on Aging and Care (SNAC): the Ministry of Health and Social Affairs, Sweden; the participating County Councils and Municipalities; and the Swedish Research Council. Specific grants were received from The Swedish Research Council for Medicine (VR; 521-2013-8676; 2017-06088; 2016-00981); the Swedish Research Council for Health, Working life and Welfare (Forte; 2016-07175; 2017-01764); Gamla Tjanarinnor (2019-00897), and the Ermenegildo Zegna Foundation. The funders had no role in study design, data collection and analysis, decision to publish, or preparation of the paper. Open access funding provided by Karolinska Institute.

## Author contributions

Conception or design of the work: D.L.V., A.R.L., A.C.L., S.F., C.V., A.M. Data analysis: A.R.L., S.F., D.L.V., A.C.L. Interpretation of the results: D.L.V., A.R.L., A.C.L., S.F., C.V., A.M., M.G.C., G.O., L.F. Drafting the article: D.L.V., A.R.L., A.C.L., A.M. Critical revision of the paper: D.L.V., A.R.L., A.C.L., S.F., C.V., A.M., M.G.C., G.O., L.F. Final approval of the paper: D.L.V., A.R.L., A.C.L., S.F., C.V., A.M., M.G.C., G.O., L.F. All the authors fulfill the ICMJE criteria for authorship.

## Competing interests

The authors declare no competing interests.
