## [Peer Review File · Nature Communications]

Reviewers' comments:

Reviewer #1 (Remarks to the Author):

This manuscript describes the results of a longitudinal population-based study on the evolution of disease clusters among patients with multimorbidity. The authors had the availability of a database that is most suited to answer their longitudinal research questions, which resulted in an interesting manuscript.

I have some questions and remarks that hopefully will help to further improve this manuscript. For a thorough critical appraisal of the statistics however, I refer to a statistical expert.

1. Since all results were presented at the patient level, what is the added value of explaining how the algorithm allows individuals to belong to more than one cluster. For clarity's sake it would at least be helpful to have the explanation of the individuals being assigned to the cluster with the highest membership score in the same paragraph.
2. I don't see the point of presenting the walking speed, disability score, MMSE etc. in table 1, since it is not used in any of the further analyses. Have the authors considered using (the evolution of) disability as a secondary outcome?
3. Authors decided to present their results in Figures 2 and 3 using column percentages. Personally, in this particular case, I would prefer row-percentages, because it allows more predictive reasoning, indicating what the risk is of belonging to a certain group. E.g., based on Figure 1, being in the group cognitive and sensory impairment at baseline give a huge (around 90%?) risk of dying during the 6 subsequent years. In my opinion the clinical relevance of this finding outreaches that of the finding
4. I find it difficult to really 'follow' the trajectories, because every point in time has different multimorbidity clusters. What does it mean when people transfer from Heart and vascular to Vascular, or from Psychiatric and respiratory to Neuropsychiatric and respiratory? What would be lost if the same clusters were used over time? Also, apparently, patients can transfer from a specific category to the unspecified group at a later moment?
5. In the discussion it is stated that there was a high heterogeneity in the multimorbidity clustering at baseline (group unspecified), which halved after 6 and 12 years of follow-up according to the authors. This is a rather optimistic interpretation: when looking at the populations still in the study at follow-up after 6 and 12 years, a similar proportion is categorized in the unspecified group. Of the 1418 subjects in the group unspecified at baseline approximately 1000 are dead or dropout after 6 years (again based on estimated numbers in Figure 1), and the proportion of people in the unspecified group remains around 40%.

Good luck with this paper, Marjan van den Akker

Reviewer #2 (Remarks to the Author):

This is an interesting paper examining longitudinal changes in empirically determined disease clusters across a 12 year period.

The authors also examine the relationship between being in a cluster and mortality.

Although well written, I had a number of concerns with the manuscript.

1. The formation of clusters is not well described. Almost 1/2 of study participants are "put" in a particular cluster at baseline. To me, this suggests that many of the clusters may contain individuals that are relatively healthy with only minor evidence of these conditions (unless the population has a very large proportion of sick 60 year olds). To me, most of the "action" is in the subjects with severe disease within the clusters.
2. The analyses are descriptive with most of them not showing statistical inference. For example, Figure 2 shows no measures of uncertainty (confidence intervals, for example). Table 1 has no measures of uncertainty either (and these are simple to calculate).
3. What was the loss to follow-up/missing data rates in this longitudinal study. The rates presented in Figure 3 assume that individuals are measured (report and are alive) at both 6 and 12 years. What proportion of the participants was this out of the total?
4. It is hard to interpret transitions of clusters when they are composed of such different diseases. For example, what does the odds ratio of dying from being in the cluster of psychiatric and respiratory disease versus unspecified group really mean? It is hard to believe that all psychiatric diseases transmit the same risk. It is even harder to believe that psychiatric diseases and respiratory diseases do.

Ref.: NCOMMS-19-04960 “Twelve-year clinical trajectories of multimorbidity in older adults: a population-based study”

Point-by-point reply to the Reviewers

Answers to Reviewer 1

1) This manuscript describes the results of a longitudinal population-based study on the evolution of disease clusters among patients with multimorbidity. The authors had the availability of a database that is most suited to answer their longitudinal research questions, which resulted in an interesting manuscript. I have some questions and remarks that hopefully will help to further improve this manuscript. For a thorough critical appraisal of the statistics however, I refer to a statistical expert.

Authors' reply

We would like to thank the Reviewer for the overall appreciation of the study and for the positive suggestions for improving the manuscript.

2) Since all results were presented at the patient level, what is the added value of explaining how the algorithm allows individuals to belong to more than one cluster. For clarity's sake it would at least be helpful to have the explanation of the individuals being assigned to the cluster with the highest membership score in the same paragraph.

Authors' reply

We thank the reviewer for this comment, which helps us to better clarify one of the novelties of our study. Presenting data on multimorbidity clusters at the individual – instead of disease – level and using a methodology that allows individuals to belong to more than one cluster is only an apparent contradiction. The use of a fuzzy cluster analysis over a hard cluster analysis helps to better handle the stochastic nature of some disease associations, the potential noise stemming from the measurement (e.g., disease assessment), and the variance due to between-individual differences. In the revised version of the manuscript we made this clearer (see methods section). Moreover, following the Reviewer's suggestion, we moved the information concerning the attribution of each individual to the cluster with the highest membership right after the explanation of the methodology (see methods section).

3) I don't see the point of presenting the walking speed, disability score, MMSE etc. in table 1, since it is not used in any of the further analyses. Have the authors considered using (the evolution of) disability as a secondary outcome?

Authors' reply

There are several reasons why we presented data on functional status of individuals even if this information was not used in further analyses. Given the well-known heterogeneity of the older population in terms of physical function and cognition, we would like to provide the readers with a comprehensive picture of the individuals included in our sample and how their functional characteristics varied across different multimorbidity patterns at baseline. Besides, this information can be useful for research purposes. For example, showing that individuals in the unspecific group had

the lowest percentage of people with slow walking speed suggests that people included in this cluster could be particularly interesting from the preventative point of view. Future studies focusing on the characteristics, and especially life styles, of this group of people could give insight into their active and healthy aging.

Regarding the question on potential analyses on the evolution of disability, we are actually planning a further manuscript on this topic. We are willing to provide preliminary data if the reviewer is interested in this subject.

4) Authors decided to present their results in Figures 2 and 3 using column percentages. Personally, in this particular case, I would prefer row-percentages, because it allows more predictive reasoning, indicating what the risk is of belonging to a certain group. E.g., based on Figure 1, being in the group cognitive and sensory impairment at baseline give a huge (around 90%?) risk of dying during the 6 subsequent years. In my opinion the clinical relevance of this finding outreaches that of the finding

Authors' reply

We agree with the Reviewer that reporting row percentages would have been more in line with a prognostic approach. However, while exploring the prognosis of the identified clusters is one of our aims (e.g., mortality analysis), looking backwards at an earlier disease status (as we do in Fig 2 and 3) helps to better speculate on the mechanisms underlying clusters' evolution over time, which is another important issue we discuss in our article. Nevertheless, we also included row percentage figures in the revised version of the manuscript. We now state in the methods section: "*Both column percentages and row percentages are provided in supplementary tables*". We also state in the results section: "*The percentages of participants moving from baseline and 6-year clusters, to 6-year and 12-year clusters, respectively, are reported in Table S4.*"

5) I find it difficult to really 'follow' the trajectories, because every point in time has different multimorbidity clusters. What does it mean when people transfer from Heart and vascular to Vascular, or from Psychiatric and respiratory to Neuropsychiatric and respiratory? What would be lost if the same clusters were used over time?

Authors' reply

We thank the Reviewer for raising this issue. One of the main findings of our study concerns precisely the dynamic nature of multimorbidity clusters. As noted by the Reviewer, the names of the clusters change over time and this is due to the fact that the individuals change their clinical status and they aggregate differently as they are affected by a different combination of diseases. We have now expanded the methods section stating: "*Such criteria were used to name multimorbidity clusters after the diseases that mostly characterized them*".

Please note that people often transit from one cluster to another for a number of reasons: 1) they develop a disease that is the consequence of another disease (e.g., psychiatric and respiratory diseases → neuropsychiatric [including dementia] and respiratory diseases); 2) people affected by more lethal diseases die allowing less severe conditions to emerge (e.g., heart and vascular disease → vascular disease); 3) iatrogenic events may trigger the development of new diseases, as for example development of psychiatric symptoms due to chronic steroid therapy (e.g., respiratory and MSK diseases → neuropsychiatric and respiratory diseases); etc. Forcing clusters to remain the same over time without accounting for their dynamic nature would have hindered us to provide a reliable picture of the clinical reality that is characterized by complexity and dynamicity. In other words, we would

have not overcome the limitations of previous works in this field, which used hard cluster analysis methods.

6) Also, apparently, patients can transfer from a specific category to the unspecified group at a later moment?

Authors' reply

First of all, we apologize for mistakenly naming one of the clusters 'unspecified' instead of 'unspecific'. We fixed this issue in the revised version of the manuscript. By using the first term, it could be easily misunderstood that many individuals could not be classified. However, they were strongly assigned to a cluster that was not characterized by any specific disease (i.e., none of the diseases were overrepresented in this group of participants). Consequently, the name for this cluster derives from the non-specific nature of diseases, and not from non-specific classification.

The unspecific cluster included people that were younger and with a better health status than those included in specific patterns at every point in time. This is due to a survival effect and explains why the large majority of people in the unspecific clusters at the 6 and 12-year follow-ups were transitioning from the unspecific cluster at baseline and 6-year follow-up, respectively. As noted by the Reviewer, a few individuals moved from a specific cluster to the unspecific cluster along time. This may be explained by the fact that the progressive accrual of new diseases and the death (or drop-out) of participants belonging to specific clusters, changed the reciprocal relation among diseases in survivors –in terms of prevalence, O/E ratio and exclusivity–, making some of them no longer classifiable into a specific cluster.

7) In the discussion it is stated that there was a high heterogeneity in the multimorbidity clustering at baseline (group unspecified), which halved after 6 and 12 years of follow-up according to the authors. This is a rather optimistic interpretation: when looking at the populations still in the study at follow-up after 6 and 12 years, a similar proportion is categorized in the unspecified group. Of the 1418 subjects in the group unspecified at baseline approximately 1000 are dead or dropout after 6 years (again based on estimated numbers in Figure 1), and the proportion of people in the unspecified group remains around 40%.

Authors' reply

In this statement, we were referring to the number of participants in the unspecific cluster –who halved after 6 and 12 years– and not to the proportion of people remaining in that cluster at each follow-up. In the revised version of the manuscript we have made it clearer in the discussion: "*The number of participants grouped in this cluster halved at 6- and 12-year follow-ups as it contributed to all the multimorbidity clusters identified at follow-ups*". Of the 1418 subjects in the unspecific group at baseline, 203 died within the first 6 years and 96 between 6-12 years, which represents in total 20% of the participants belonging to the unspecific group at baseline. Other clusters presented a much higher mortality.

Answers to Reviewer 2

1) This is an interesting paper examining longitudinal changes in empirically determined disease clusters across a 12 year period. The authors also examine the relationship between being in a cluster and mortality.

Although well written, I had a number of concerns with the manuscript.

The formation of clusters is not well described. Almost 1/2 of study participants are "put" in a particular cluster at baseline. To me, this suggests that many of the clusters may contain individuals that are relatively healthy with only minor evidence of these conditions (unless the population has a very large proportion of sick 60 year olds). To me, most of the "action" is in the subjects with severe disease within the clusters.

Authors' reply

Given that our analytical approach represents a novelty in the field of multimorbidity and in longitudinal studies, we did our best to provide a clear description of the methods. Thanks to the comments of the two Reviewers, in the revised version of the manuscript we tried to be more specific regarding the way clusters have been built and named (see tracked changes in the methods section).

In our study, the identification of clusters of individuals is based on the underlying patterns and combinations of diseases. As noted by the Reviewer, 52% of the baseline participants were included in a specific cluster where specific diseases were clearly overrepresented, and 48% of them were grouped in an unspecific cluster where none of the diseases considered were overrepresented.

We agree with the Reviewer that specific groups of diseases may play a major role due to their severity, and the lack of information related to disease severity was indeed accounted for as a limitation of our study. In the revised version of the manuscript, we further expand the discussion by stating: *“Disease severity may indeed partially explain the clinical trajectories described in the present study. Still, in our opinion, independently from disease severity, the insights on the natural evolution of multimorbidity provided in this study are highly valuable and cover an important knowledge gap left by previous cross-sectional studies. Moreover, there is evidence that the burden of specific conditions changes depending on the comorbidity status of one individual, making it difficult – especially in older individuals – to reliably assess disease severity. Finally, we provided information on drug utilization, physical function, and cognitive function, which somehow reflect disease severity”*.

Despite this limitation, in our study we investigate (and this was one of the major aims of the study) which are the multimorbidity clusters characterized by worst prognosis. As shown in table 2, in our sample, the multimorbidity clusters characterized by highest mortality were those presenting an exceptionally high prevalence of cardiovascular, neuropsychiatric and respiratory diseases. Finally, in a previous study we demonstrated that the association between specific multimorbidity clusters and negative outcomes might –to a certain extent– be independent of specific diseases and their severity (e.g., see supplementary analysis of Vetrano DL et al; PlosMed 2018), suggesting a group-specific prognostic value of a given multimorbidity cluster.

2) The analyses are descriptive with most of them not showing statistical inference. For example, Figure 2 shows no measures of uncertainty (confidence intervals, for example). Table 1 has no measures of uncertainty either (and these are simple to calculate).

Authors' reply

Indeed, not reporting measures of uncertainty and variance (i.e., confidence intervals and standard deviations) in our descriptive analyses was a deliberate choice, in order to avoid too crowded tables and figures. We are of course able to provide any kind of additional statistics for each of the reported parameters, which depending on the Editor's indications, we can report either in the manuscript or in supplementary tables and figures. To note, inferential analyses on mortality are already reported with odds ratios and 95% confidence intervals.

3) What was the loss to follow-up/missing data rates in this longitudinal study. The rates presented in Figure 3 assume that individuals are measured (report and are alive) at both 6 and 12 years. What proportion of the participants was this out of the total?

Authors' reply

We do not have missing data on diseases (which would have been a major issue for the robustness of the cluster analysis) neither at baseline nor follow-ups, which is the reason why this information is not provided. However, we agree with the Reviewer that the proportion of drop-outs and dead participants reported in figure 2 and figure 3 do not refer to the total sample, but to the 1716 and 1016 individuals alive at the 6- and 12-year follow-ups, respectively. When the whole sample population is considered (N=2931), 28% of the participants died and 14% dropped out between 0-6 years, and 16% died and 8% dropped out between 6-12 years. In the results of the revised version of the manuscript we state that: *“Over the 12 years, 1290 (44%) deaths occurred (28% within the first 6 years and 16% between 6 and 12 years). Moreover, 625 (22%) individuals dropped out (14% within the first 6 years and 8% between 6 and 12 years).”*

4) It is hard to interpret transitions of clusters when they are composed of such different diseases. For example, what does the odds ratio of dying from being in the cluster of psychiatric and respiratory disease versus unspecified group really mean? It is hard to believe that all psychiatric diseases transmit the same risk. It is even harder to believe that psychiatric diseases and respiratory diseases do.

Authors' reply

This is a relevant issue. Indeed, complexity characterizes disease status of most of the older adults, who in our sample are, on average, affected by more than five different diseases. Such diseases have the potential to interact within the same individual in an endless number of different combinations. Our study aims precisely to describe the disease patterns according to which individuals can –to a certain extent– be considered part of homogeneous groups.

In older and complex people, a single-disease approach at any stage of the care path –from prevention to end of life– has proved inadequate and far from the real needs of these individuals. In the era of personalized medicine, identifying individual-level characteristics on which we may leverage through targeted and effective interventions is of utmost importance. However, when it comes to complex older adults, identifying groups of people –instead of single individuals– that may benefit from a given intervention might be considered a more reasonable and accomplishable goal. In this regard, multimorbidity clusters may be thought of as indicators of the overall morbidity burden affecting one or more body systems and, eventually, as a mean to facilitate targeted interventions.

In the discussion of our manuscript, under the heading “Multimorbidity clusters and trajectories”, we identify and discuss some of the potential pathophysiological pathways that can explain most of the transitions among multimorbidity clusters observed during 12 years. Considered this issue, and backed up by an extensive literature, we do not completely agree with the Reviewer's concerns. We do not support the idea that all psychiatric diseases or all cardiovascular diseases transmit the same mortality

risk. What we did in this study, among other results, was to report the association with mortality of specific groups of individuals, sharing the same underlying patterns of chronic diseases.

Reviewers' comments:

Reviewer #3 (Remarks to the Author):

I enjoyed reading this paper. However, there are some minor details I would like to address in continuation of the existing review.

Regarding questions from previous reviewer 2:

The clusters are sufficiently described in the present version. One thing that is not 100% clear to me, is how the fuzzy classification translates into the figures: Are patients present in multiple of the clusters (i.e. patients can be present in multiple clusters in the figures) or are patients associated to exactly one cluster for the figure.

I am convinced that a thorough job was carried out in selecting the correct parameters. However, the statistics of the 100 repetitions and 2 times cross-validations are not included in the paper. Including these will give more statistical inference besides what was added in table 1 by request in previous review.

Regarding mortality, the additions make sense. However, the age of onset in the groups can also explain the difference: "Cognitive and sensory impairment" has the highest mean age (88.2) for the baseline and the "Heart diseases and cognitive impairment" has the highest mean age (85.5) for the 6 years stop. It is not too surprising that they have the highest number of people who die in the next 6-year period. The discussion does not mention this and does not mention the logistic regression that was carried out to test the significance of the large mortality (table 2). The p-values and log-odds from table 2 should also be mentioned somewhere in the text. It would also clarify things, if the method section mentions the setup of the logistic regression (that it corrects for age, gender and education and that the comparison group is the unspecific group). Mentioning that X tests were performed and corrected for multiple testing (I assume) along with some short summary (e.g. number of significant tests) in the results section would also be nice.

It is not intuitive for me to see what the purpose of figure 1 and 2 are. I would suggest adding something like "from" in front of "baseline" and "to" in front of "6 years" (figure 1).

I agree that figure 1 is difficult to read. One way of simplifying it, would be to filter out transitions where there are few patients. The authors could consider doing this.

Although I feel it is not a prerequisite for publication, I will note that the results of this study could be compared to two other studies: 1) A comorbidity study of a Swedish population (citation below) and 2) the study cited as reference 21. The first study (1) report comorbidities that can be compared to the clusters. The second study (2) reports the transitions of one diagnosis to another. This could be compared to the transitions found in this paper. Although neither of these methods have the same approach as the current paper, they can still show some agreement.

Citation for (1): Tanushi, H., Dalianis, H. & Nilsson, G. H. Calculating Prevalence of Comorbidity and Comorbidity Combinations with Diabetes in Hospital Care in Sweden Using a Health Care Record Database. In: Proceedings of LOUHI 2011 Third International Workshop on Health Document Text Mining and Information Analysis 59–65 (Bled, Slovenia, 2011)

<https://www2.dsv.su.se/comorbidityview-demo/>

General:

While the paper is written in a very nice language, it does not conform to the results-then-methods way of writing. The result section has simply been put in from of the method section without any attempt at making it readable without the methods. The result section needs: 1) introduction to the data size 1.a) the exclusion of the 432 patients and the significance associated with them being male is more a result than method in any case 2) definition of the key concepts like O/E abbreviates, description of what input and parameters was used for the fuzzy classification, etc. Therefore, a significant portion of the methods sections must be moved to results.

Ref.: NCOMMS-19-04960 “Twelve-year clinical trajectories of multimorbidity in older adults: a population-based study”

Point-by-point reply to the Reviewers

Answers to Reviewer #3

I enjoyed reading this paper. However, there are some minor details I would like to address in continuation of the existing review.

Reviewer’s comment: Regarding questions from previous reviewer 2: The clusters are sufficiently described in the present version. One thing that is not 100% clear to me, is how the fuzzy classification translates into the figures: Are patients present in multiple of the clusters (i.e. patients can be present in multiple clusters in the figures) or are patients associated to exactly one cluster for the figure.

Authors’ reply: We thank the Reviewer for underlying the need to clarify this point. For the descriptive and mortality analyses we necessarily needed to deal with discrete categories. Consequently, we forced the study participants into the cluster they were more likely to belong to according to the membership matrix derived from the fuzzy c-means cluster analysis. In the revised version of the manuscript we specify in the methods: “*Shifts between clusters were computed by cross-tabulating individuals between each wave (baseline to 6-year follow-up and 6-year to 12-year follow-up) after assigning them to the cluster where they were more likely to belong. In this way, we analyzed the most likely individual trajectories.*” Moreover: “*Logistic regression models were fitted to estimate the association between clusters and mortality. Also in this case, participants were assigned to the cluster where they were more likely to belong.*” Finally, the following note has been added to Figures 1, 2, 3 and Tables 1 and 2: “*To note, for this analysis participants were assigned to the cluster where they were more likely to belong.*”

Reviewer’s comment: I am convinced that a thorough job was carried out in selecting the correct parameters. However, the statistics of the 100 repetitions and 2 times cross-validations are not included in the paper. Including these will give more statistical inference besides what was added in table 1 by request in previous review.

Authors’ reply: We thank the Reviewer for this helpful suggestion. We now provided this information. In the revised version of the results we stated: “*Solutions were evaluated based on their clinical consistency and significance criteria (Figures S1-S15).*” Moreover, in the revised version of the methods section we specify: “*The main parameters used during our cluster analysis were the number of clusters and a fuzziness parameter, denoted as “m”, which ranges from just above 1 to infinity. High m values produce a fuzzy set of c clusters, so that individuals are equally distributed across clusters, whereas lower ones generate non-overlapped clusters. In our study we checked m=1.1, 1.2, 1.4, 1.5, 2, 4 over 1 to 20 cluster combinations; the value m=1.1 over performed the rest of values. Since clustering algorithms are unsupervised techniques, the model fitting the dataset is traditionally computed through cost functions that depend on both the dataset and the clustering parameters and are denoted as validation indices. We computed different validation indices to obtain the optimal number of clusters c and the optimal value of the fuzziness parameter m. Included parameters were: the Fukuyama index (optimal when presenting low values), the Xie-Beni index (optimal when presenting low values), the Partition coefficient index (optimal when presenting high values), the Partition entropy index (optimal when presenting low values), and the Calinski-Harabasz index (optimal when presenting high values; Figures S1-S15).*”

Reviewer’s comment: Regarding mortality, the additions make sense. However, the age of onset in the groups can also explain the difference: “Cognitive and sensory impairment” has the highest mean age (88.2) for the baseline and the “Heart diseases and cognitive impairment” has the highest mean age (85.5) for the 6 years stop. It is not too surprising that they have the highest number of people who die in the next 6-year period. The discussion does not mention this and does not mention the logistic regression that was carried out to test the significance of the large mortality (table 2). The p-values and log-odds from table 2 should also be mentioned somewhere in the text. It would also clarify things, if the method section mentions the setup of the logistic regression (that it corrects for age, gender and education and that the comparison group is the unspecific group). Mentioning that X tests were performed and corrected for multiple testing (I assume) along with some short summary (e.g. number of significant tests) in the results section would also be nice.

Authors’ reply: We are glad to clarify this issue. It is true that participants in the *cognitive and sensory impairment* cluster at baseline, and in the *heart diseases and cognitive impairment* cluster at the 6-year follow-up are the oldest and present a higher crude mortality (see Table 2; event/at risk columns). However, the adjustment carried out in the logistic regression is intended precisely to account for individuals’ characteristics that may bias the estimated association. In fact, upon adjustment by age, sex and education, the *heart diseases and cognitive impairment* cluster at the 6-year follow-up is no longer associated with a higher mortality, but the *neuropsychiatric and respiratory diseases* cluster is. For a better interpretation of the results, in the revised Table 2 we included the cumulative death incidence (%) in the events/at risk columns. When in the discussion we mention the clusters including cardiovascular and neuropsychiatric diseases as those most strongly associated with mortality, we refer to the cumulative impact of clusters including such diseases (i.e., at 6 years there are two clusters characterized by heart diseases). To make it clearer, in the revised version of the discussion we specify: “...Those clusters accounted for 51% of deaths during the first follow-up and for 57% of deaths during the second follow-up. Notably, at six years there were two clusters characterized by cardiovascular diseases.”

Moreover, in the revised version of the discussion we mention the results from the logistic analysis: “At least four out of ten participants died over the course of the study. Both at baseline and at six year follow-up, individuals with multimorbidity patterns characterized by cardiovascular and neuropsychiatric diseases had the highest mortality; with multi-adjusted odds ratios ranging between 1.60 and 6.00 (taking people in the unspecific cluster as the reference).”

Following the Reviewer’s suggestion we reported in the results section the measures of association between the clusters and mortality: “As shown in **table 2**, at baseline the heart diseases (OR 3.07; 95% CI 2.26–4.19), the cognitive and sensory impairment (OR 6.00; 95% CI 4.21–8.54), and the psychiatric and respiratory diseases (OR 1.60; 95% CI 1.02–2.51) clusters were significantly associated with a higher six-year mortality, compared with people in the unspecific cluster. These clusters accounted for 51% of deaths. At first follow-up, the heart and vascular diseases (OR 3.78; 95% CI 2.13–6.70), the heart diseases and cognitive impairment (OR 3.73; 95% CI 2.41–5.79), and the neuropsychiatric and respiratory diseases (OR 4.30; 95% CI 2.95–6.27) clusters had the highest OR for six-year mortality, compared with the group of people in the unspecific cluster. These clusters accounted for 57% of deaths in the following six years.”

In the revised version of the methods section we specify: “Logistic regression models adjusted by age, sex and education were fitted to estimate the association between clusters and mortality, using the unspecific cluster as the reference group.”

Regarding the multiple testing, both overall and pairwise tests were performed but we did not report the p-values as the vast majority of overall tests were highly significant (see table below).

p-value	Baseline tests (%)	6-years tests (%)	12-years tests(%)
<0.001	60.0	42.8	36.7
<0.01	7.8	10.0	7.2

<0.05	5.6	12.8	6.7
>=0.05	26.7	34.4	49.4

Percentage of performed pairwise test vs p-value

In the revised version of the methods section we report: *“Although the overall number of significant tests among clusters remained stable at each follow up, the number of highly significant pairwise statistical test (i.e., $p < 0.001$) decreased from 60.0% to 36.7%”.*

Reviewer’s comment: It is not intuitive for me to see what the purpose of figure 1 and 2 are. I would suggest adding something like “from” in front of “baseline” and “to” in front of “6 years” (figure 1).

Authors’ reply: We appreciate this suggestion very much, which helps us to improve the interpretation of figures 2 and 3. In the revised version of such figures we changed the labels to: “from baseline” and “at 6 years” (figure 2), and “from 6 years” and “at 12 years” (figure 3). We prefer to use “at” given that the numbers are column percentages.

Reviewer’s comment: I agree that figure 1 is difficult to read. One way of simplifying it, would be to filter out transitions where there are few patients. The authors could consider doing this.

Authors’ reply: We understand the impression of the Reviewer; however, after a thoughtful group discussion, for the sake of transparency and given that it would be difficult to choose a threshold to omit some of the transitions, we prefer to stick to the current version of figure 1.

Reviewer’s comment: Although I feel it is not a prerequisite for publication, I will note that the results of this study could be compared to two other studies: 1) A comorbidity study of a Swedish population (citation below) and 2) the study cited as reference 21. The first study (1) report comorbidities that can be compared to the clusters. The second study (2) reports the transitions of one diagnosis to another. This could be compared to the transitions found in this paper. Although neither of these methods have the same approach as the current paper, they can still show some agreement.

Citation for (1): Tanushi, H., Dalianis, H. & Nilsson, G. H. Calculating Prevalence of Comorbidity and Comorbidity Combinations with Diabetes in Hospital Care in Sweden Using a Health Care Record Database. In: Proceedings of LOUHI 2011 Third International Workshop on Health Document Text Mining and Information Analysis 59–65 (Bled, Slovenia, 2011)

Authors’ reply: We thank the Reviewer for these suggestions. The report from Tanushi and Colleagues, despite describing the interesting relationship between diabetes and its comorbidities using administrative Swedish data, it does not report longitudinal findings, which limits the comparability with our results.

Regarding the study from Jensen and Colleagues (ref #21), in the discussion section we wrote: *“Another study of a population-wide registry of more than six million patients in Denmark showed more than a thousand significant longitudinal disease trajectories and some major multimorbidity clusters characterized by diseases of the prostate, chronic obstructive pulmonary disease, cerebrovascular disease, cardiovascular disease, and diabetes mellitus. The study had the limitation of data drawn retrospectively from a hospital registry of primary and secondary diagnostic codes. Further, both chronic and acute diseases were included.”*

Reviewer’s comment: While the paper is written in a very nice language, it does not confirm to the results-then-methods way of writing. The result section has simply been put in from of the method section without any attempt at making it readable without the methods. The result section needs: 1) introduction to the data size 1.a) the exclusion of the 432 patients and the significance associated with them being male is more a result than method in any case 2) definition of the key concepts like O/E

abbreviates, description of what input and parameters was used for the fuzzy classification, etc. Therefore, a significant portion of the methods sections must be moved to results.

Authors' reply: As recommended by the Reviewer, in order to adapt the manuscript to a results-then-method style, we propose a new version of the results section where we include details about the methodology used and new headlines summarizing the main message of each paragraph:

Results

Six clusters of individuals with multimorbidity were identified. The present study is based on data from the Swedish National Study on Aging and Care in Kungsholmen (SNAC-K), an ongoing population-based study started in 2001 and involving 3363 individuals aged ≥ 60 years from a central area in Stockholm, Sweden. From the whole sample, 432 participants with less than 2 chronic disease have been excluded (i.e., those without multimorbidity). Those excluded were younger, reported a higher level of education, and were more often male than those included in the study ($p < 0.001$). At baseline, study participants' mean age was 76.1 ± 11.0 and 1951 (66.6%) were female. Over the 12 years, 1290 (44%) deaths occurred (28% within the first 6 years and 16% between 6 and 12 years). Moreover, 625 (22%) individuals dropped out (14% within the first 6 years and 8% between 6 and 12 years). At each follow-up, we performed a dimensionality reduction (i.e., multiple correspondence analysis) to obtain the input data for participants' clustering. A fuzzy c-means cluster analysis with optimal a fuzziness parameter at $m=1.1$ (which outperformed other m values; see methods) was employed to identify clusters of individuals based on their underlying patterns of multimorbidity. Using an observed/expected ratio ≥ 2 (O/E ratio; i.e., the ratio between the prevalence of a given condition in a cluster and its prevalence in the whole sample) and an exclusivity $\geq 25\%$ (i.e., the proportion of individuals with a given condition in the whole sample that belong to a cluster) for each disease, five clusters of people were identified at baseline: those with *psychiatric and respiratory diseases* (5.4%), *heart diseases* (9.3%), *respiratory and musculoskeletal diseases* (15.7%), *cognitive and sensory impairment* (10.6%), and *eye diseases and cancer* (10.7%; **table S3**). Solutions were evaluated based on their clinical consistency and significance criteria (**Figures S1-S15**). Half of the people (48.7%) were grouped in an additional *unspecific* cluster, as they were affected by prevalent diseases but whose occurrence did not exceed the expected. Similarly, five clusters (plus the unspecific one) were identified at 6 and 12 years. At follow-ups, those diseases characterizing the baseline clusters were regrouped into different multimorbidity clusters. The clinical characteristics of the clusters are reported in **table 1**.

Individuals had different demographic, clinical and functional profiles across the clusters.

Descriptive analyses were carried out to characterize the six clusters of individuals with multimorbidity. At baseline, participants in the *cognitive and sensory diseases*, the *eye diseases and cancer*, and the *heart diseases* clusters were the oldest. Participants in the *heart diseases*, the *eye diseases and cancer*, and the *psychiatric and respiratory diseases* clusters presented the greatest number of chronic diseases (mean number: 7.7 ± 2.4 [standard deviation], 6.0 ± 2.0 , and 5.7 ± 2.2 , respectively). Participants in the *heart diseases* and *psychiatric and respiratory diseases* clusters and those in the *cognitive and sensory impairment* cluster used the highest number of drugs (mean number: 7.7 ± 3.5 , 6.2 ± 3.7 , and 6.1 ± 3.4 , respectively). Moreover, individuals included in the *heart diseases*, the *eye diseases and cancer*, and the *cognitive and sensory impairment* clusters presented the highest prevalence of disability and slow walking speed. The *cognitive and sensory impairment* and the *psychiatric and respiratory diseases* cluster showed the lowest MMSE scores. The *unspecific* cluster was characterized by the lowest mean age and the lowest number of chronic diseases and drugs. This group had the lowest prevalence of disability and the highest walking speed, yet it had a high prevalence of hypertension, diabetes, dyslipidemia, and obesity. Such conditions were frequent also among participants in the *heart diseases* and the *eye diseases and cancer* clusters.

At follow-ups, in spite of varied clustering, a similar clinical distribution was observed for the different types of disorders. That is, people in clusters characterized by cardiovascular, neuropsychiatric, and respiratory diseases showed the highest number of diseases and drugs and the highest levels of functional impairment.

Patterns of transitions between clusters can be identified over time. Upon assigning individuals to the cluster they were more likely to belong to, we described their trajectories as they moved between clusters or to death over time. **Figure 1** depicts the longitudinal evolution of multimorbidity clusters over 12 years and includes both the change over time of disease patterns (the diseases that characterize a specific cluster of individuals) and the migration of participants from one cluster to another. The height of the boxes and the thickness of the stripes in the figure are proportional to the amounts of people in the cluster and moving out from the cluster, respectively.

In order to better characterize such transitions, we report in **figure 2** and **figure 3** the proportion of participants that were part of the 6-year and 12-year follow-ups clusters and that moved from multimorbidity clusters detected at an earlier wave. The percentages of participants moving from baseline and 6-year clusters, to 6-year and 12-year clusters, respectively, are reported in Tables S4a, S4b, S4c, and S4d. During both first and second follow-up periods, the main shifts among clusters involved participants in the *unspecific* cluster, who moved primarily to clusters characterized by cardiovascular, eye, respiratory and musculoskeletal diseases. For example, persons in the *unspecific* group at baseline moved and represented 48.7%, 45.0% and 38.8% of the 6-year follow-up *heart and vascular diseases*, *musculoskeletal, respiratory and immune diseases*, and *eye diseases* clusters, respectively. Similarly, persons belonging to the *unspecific* group at the 6-year follow-up moved and represented 49.5%, 49.1% and 20.6% of the 12-year follow up *cardiometabolic diseases*, *eye and musculoskeletal diseases*, and *vascular diseases* clusters, respectively.

Different multimorbidity clusters confer different mortality risks. The association between the multimorbidity clusters and mortality was tested in logistic regression models adjusted by age, sex and education, taking the *unspecific* cluster as the reference group. As shown in **table 2**, at baseline the *heart diseases* (OR 3.07; 95% CI 2.26–4.19), the *cognitive and sensory impairment* (OR 6.00; 95% CI 4.21–8.54), and the *psychiatric and respiratory diseases* (OR 1.60; 95% CI 1.02–2.51) clusters were significantly associated with a higher six-year mortality, compared with people in the *unspecific* cluster. These clusters accounted for 51% of deaths. At first follow-up, the *heart and vascular diseases* (OR 3.78; 95% CI 2.13–6.70), the *heart diseases and cognitive impairment* (OR 3.73; 95% CI 2.41–5.79), and the *neuropsychiatric and respiratory diseases* (OR 4.30; 95% CI 2.95–6.27) clusters had the highest OR for six-year mortality, compared with the group of people in the *unspecific* cluster. These clusters accounted for 57% of deaths in the following six years.

REVIEWERS' COMMENTS:

Reviewer #3 (Remarks to the Author):

All issues I commented on have been dealt with. I found three typos while reading the paper again. I trust that my input is not needed again for any of these corrections.

Page 25:

the Xie-Beni index (optimal when presenting lo values),

“lo” -> “low”

Page 25:

... Calinski-Harabasz index (optimal when presenting high values; Figures S1-S15). [44]. Given ...

Reference 44 should not be in a sentence by itself

Also: Do you have single references for each of the measures? It would be nice to have them here rather than have to look them up.

Page 26:

Tukey method were used when explanatory variable was normal-distributed or Benjamini & Hochberg (Benjamini and Hochberg 1995) method otherwise.

Correction suggestion (marked with *): “.. when *the* explanatory variable was normal-distributed and Benjamini & Hochberg *[reference number]* method otherwise.”

REVIEWERS' COMMENTS:

Reviewer #3 (Remarks to the Author):

All issues I commented on have been dealt with. I found three typos while reading the paper again. I trust that my input is not needed again for any of these corrections.

Page 25:

the Xie-Beni index (optimal when presenting lo values),
“lo” -> “low”

Authors' reply: We fixed this typo according to the Reviewer's suggestion.

Page 25:

... Calinski-Harabasz index (optimal when presenting high values; Figures S1-S15). [44]. Given ...
Reference 44 should not be in a sentence by itself

Also: Do you have single references for each of the measures? It would be nice to have them here rather than have to look them up.

Authors' reply: We fixed this according to the Reviewer's suggestion.

Page 26:

Tukey method were used when explanatory variable was normal-distributed or Benjamini & Hochberg (Benjamini and Hochberg 1995) method otherwise.

Correction suggestion (marked with *): “.. when *the* explanatory variable was normal-distributed and Benjamini & Hochberg *[reference number]* method otherwise.”

Authors' reply: We changed this following the Reviewer's suggestion.